# Morphology, Phenotype, and Molecular Identification of Clinical and Environmental *Fusarium solani* Species Complex Isolates from Malaysia

**DOI:** 10.3390/jof8080845

**Published:** 2022-08-11

**Authors:** Jasper E. James, Jacinta Santhanam, Latiffah Zakaria, Nuraini Mamat Rusli, Mariahyati Abu Bakar, Satinee Suetrong, Jariya Sakayaroj, Mohd Fuat Abdul Razak, Erwin Lamping, Richard D. Cannon

**Affiliations:** 1Biomedical Science Programme, Faculty of Health Sciences, Universiti Kebangsaan Malaysia, Kuala Lumpur 50300, Malaysia; 2School of Biological Sciences, Universiti Sains Malaysia, Gelugor 11800, Malaysia; 3National Center for Genetic Engineering and Biotechnology (BIOTEC), Thailand Science Park, Pathum Thani 12120, Thailand; 4School of Science, Wailalak University, Nakhonsithammarat 80161, Thailand; 5Bacteriology Unit, Institute for Medical Research, National Institute of Health, Shah Alam 40170, Malaysia; 6Sir John Walsh Research Institute, Faculty of Dentistry, University of Otago, Dunedin 9016, New Zealand

**Keywords:** *Fusarium solani*, FSSC, MLST, *TEF1-α*, *RPB2*

## Abstract

*Fusarium* infections in humans (fusariosis) and in economically important plants involve species of several *Fusarium* species complexes. Species of the *Fusarium solani* species complex (FSSC) are the most frequent cause of human fusariosis. The FSSC comprises more than 60 closely related species that can be separated into three major clades by multi-locus sequence typing (MLST) using translation elongation factor 1-alpha (*TEF1-α*) and RNA polymerase II (*RPB2*) DNA sequences. The MLST nomenclature for clade 3 of the FSSC assigns numbers to species types (e.g., FSSC 2) and lowercase letters to identify unique haplotypes. The aim of this study was to analyse the genotypic and phenotypic characteristics of 15 environmental and 15 clinical FSSC isolates from Malaysia. MLST was used for the genotypic characterisation of FSSC isolates from various locations within Malaysia, which was complemented by their morphological characterisation on potato dextrose and carnation leaf agar. MLST identified eight different FSSC species: thirteen *Fusarium keratoplasticum* (i.e., FSSC 2), six *Fusarium suttonianum* (FSSC 20), five *Fusarium falciforme* (FSSC 3+4), two *Fusarium cyanescens* (FSSC 27)*,* and one each of *Fusarium petroliphilum* (FSSC 1), *Fusarium waltergamsii* (FSSC 7), *Fusarium* sp. (FSSC 12), and *Fusarium striatum* (FSSC 21). Consistent with previous reports from Malaysia, most (11 of 15) clinical FSSC isolates were *F. keratoplasticum* and the majority (9 of 15) of environmental isolates were *F. suttonianum* (5) or *F. falciforme* (4) strains. The taxonomic relationships of the isolates were resolved phylogenetically. The eight *Fusarium* species also showed distinct morphological characteristics, but these were less clearly defined and reached across species boundaries. Although *TEF1-α* and *RPB2* sequences were sufficient for the species identification of most FSSC isolates, a more precise MLST scheme needs to be established to reliably assign individual isolates of the species-rich FSSC to their geographically-, epidemiologically-, and host-associated sub-lineages.

## 1. Introduction

*Fusarium*, a genetically complex genus, belongs to the fungal order Hypocreales, in the class Sordariomycetes. The fusaria are globally distributed in environments such as soil, water, plants, and human habitats. Fusaria are significant plant pathogens causing severe vascular wilt and root rot disease in agriculturally important crops [1,2,3]. The *Fusarium* head blight of wheat [4] and *Fusarium* wilt of bananas [5] are amongst the most devastating diseases in plant hosts, causing heavy losses in global food production with an enormous impact on the communities that depend on these crops [6,7]. *Fusarium* wilt of banana, also known as Panama disease [8], has caused several devastating crop losses to the global banana industry [9,10].

Fusaria are also among the most frequent causes of invasive mould infections (IMIs), second only to aspergilli [11]. The increasing number of immunocompromised patients [12,13] heightens the concern for IMIs as they typically have high mortality rates (56%) [14], especially in neutropenic patients where the mortality rate reaches as high as 100% [15]. Fusaria comprise 20 species complexes [16]. The *Fusarium solani* species complex (FSSC) containing more than 60 different species accounts for ~60% of fusariosis cases worldwide [17,18], ranging from localised skin, nail, and eye infections to life-threatening disseminated IMIs [19].

Historically, species of the FSSC were simply referred to as *Fusarium solani* based on morphological features, host-specific pathogenicity, and sexual/asexual compatibility [20]. However, a *Fusarium* keratitis outbreak in the United States [21] changed that approach through the application of multi-locus sequence typing (MLST). FSSC isolates separated into three distinct clades based on the ribosomal RNA (rRNA) internal transcribed spacer (ITS) region, the D1 and D2 domains of the nuclear large-subunit (LSU) rRNA genes, translation elongation factor 1-alpha (*TEF1-α*), and RNA polymerase II (*RPB2*) sequences [22]. Clade 1 and clade 2 species of the FSSC are geographically restricted and exclusively associated with plants. They appear to be endemic to New Zealand and South America, respectively [23]. Members of FSSC clade 3 are more common in highly populated areas, and they seem to grow faster and produce more conidia than members of clades 1 or 2 [24]. The informal haplotype nomenclature of FSSC clade 3 species [21,22] was developed to facilitate sharing of more accurate epidemiological data by assigning individual isolates to a particular MLST. For example, FSSC clade 3 isolates with the MLST 1-b or 2-a are *F. petroliphilum* (FSSC 1) or *F. keratoplasticum* (FSSC 2) isolates, respectively. In 2018, Sandoval-Denis et al. proposed renaming species of the FSSC as *Neocosmospora* [25,26]. This proposal was, however, subsequently rebutted [27], and there is no final decision as to which nomenclature is more accurate or useful in a clinical setting [28].

Infections by species of the FSSC are problematic because most are innately resistant to mainstream azole antifungals, including voriconazole [29,30], which is the recommended treatment option for invasive fusariosis (IF) [31,32]. Amphotericin B seems to be the only antifungal to which most FSSC species are susceptible [29,30]. This hinders the efficacy of treating immunocompromised patients with IF [33,34,35,36]. The widespread use of fungicides in agriculture [37] has led to the rise and global spread of azole resistant *Aspergillus fumigatus* isolates [38] into hospital settings [39,40,41], some of which are beginning to exhibit resistance to multiple classes of antifungals [42]. A similar hypothesis was recently put forward for the emergence of *Candida auris*, a drug resistant human fungal pathogen of serious clinical concern. An investigation by Yadav et al. [43], provided strong evidence for the possible emergence and spread of multidrug resistant *C. auris* clinical isolates, usually found in marine habitats, through the application of fungicides on stored apples. A detailed review of the concerning trend for the selection of drug resistant fungal pathogens, including various *Candida* species and *Cryptococcus neoformans*, through the extensive use of agrochemicals in agriculture and the wood industry is provided by Bastos et al. [44].

The azole target, lanosterol-14α-demethylase, encoded by *CYP51* genes [45], has three orthologues in *Fusarium*: *CYP51A*, *CYP51B*, and *CYP51C* [46]. We have recently reported a strong association between a *CYP51A* promoter deletion and voriconazole resistance in a collection of 25 Malaysian FSSC isolates, 20 of which were included in this study [46]. The 23 bp *CYP51A* promoter deletion was present in all voriconazole resistant FSSC isolates (six *F. keratoplasticum* and three *F. suttonianum* isolates) that could be sequenced. Unfortunately, the presence of the 23 bp *CYP51A* promoter deletion could not be confirmed for two additional voriconazole resistant clinical *F. falciforme* isolates (Ff541 and Ff0020), because neither DNA oligomer primer pair used could amplify this region for DNA sequencing. Interestingly, however, this region could be amplified from the remaining three voriconazole sensitive environmental *F. falciforme* isolates. As expected, none of these three isolates contained the 23 bp *CYP51A* promoter deletion [46]. These findings indicated a possible DNA exchange of the 23 bp *CYP51A* promoter deletion between the ancestors of the voriconazole resistant *F. keratoplasticum* and *F. suttonianum* and possibly also *F. falciforme* isolates via asexual recombination and positive selection by azole fungicide use in agriculture [46,47,48]. In the *Fusarium fujikuroi* species complex [49], however, antifungal resistance appears to be species specific [1]. Thus, a carefully designed MLST scheme for the identification of individual *Fusarium* isolates is essential to determine appropriate antifungal treatment therapy.

In Malaysia, most reports of *Fusarium* infections are related to plant infections of oil palm [50], pineapple [51], or paddy field soils [52], while another study reported that sea turtles were infected with *Fusarium* [53]. A study of 1449 environmental *Fusarium* isolates from the Malaysian highlands found that 66.1% belonged to the FSSC [54], highlighting its predominance in Malaysia. The first fusariosis case in Malaysia, a patient diagnosed with *Fusarium* keratitis, was reported in 1981 [55], and a 5-year retrospective review (2007–2011) of fungal keratitis at the Universiti Sains Malaysia Hospital found that almost half (46%; 19 of 41) of all fungal keratitis cases were caused by FSSC species [56].

The present study aimed to determine the phenotypic and genotypic relationships of FSSC isolates and their species distribution in the environment and in clinical samples. We hypothesized that genotypic identification through a molecular approach is more accurate and informative in a clinical context, as the nature of antifungal resistance in fusaria is species-specific and the taxonomic clade of the FSSC is diverse. The comparison between phenotypic and genotypic characteristics is an important first step to enable accurate and rapid identification of isolates of the FSSC so that effective antifungal treatment can be implemented for *Fusarium* infections of humans, animals, and plants. This study describes the species distribution of 30 FSSC isolates collected in Malaysia, and it provides a detailed phenotypic and morphological characterization of 10 clinical and 15 environmental isolates from that collection.

## 2. Materials and Methods

### 2.1. Fungal Isolates

The 15 clinical *Fusarium* isolates were obtained from the Hospital Canselor Tuanku Muhriz UKM and the Institute for Medical Research, Malaysia. They had originally been collected from nail, skin, corneal scrapings, and blood as a part of routine diagnostic procedures (Table 1). No identifying data from any patients were obtained or utilised in this study. A further 15 environmental isolates were from soil and plant debris located across six states in Peninsular Malaysia: Terengganu, Pahang, Kelantan, Perak, Kedah, and Selangor (Table 1). All isolates were presumptively identified as FSSC species based on their conidia morphology [57]. The antifungal susceptibilities of 25 of the 30 isolates to itraconazole/posaconazole, voriconazole, and amphotericin B were previously reported [46].

### 2.2. Morphological Examination

Macromorphological examinations were conducted on isolates grown on potato dextrose agar (PDA; Merck, Kenilworth, NJ, USA) plates incubated at 28 °C for fourteen days. The diameter and the colour of the colony surface and substrate mycelia (reverse side/bottom of agar plate) were recorded. For microscopic examination, each isolate was sub-cultured on approximately 1 cm^2^ blocks of agar containing a piece (0.5 cm^2^) of sterile carnation leaf agar (CLA). The agar blocks were placed on a sterile microscope slide, covered with a sterile cover slip, and incubated inside a petri dish for 12 h periods of light (day) and 12 h periods in the dark (night) by using a 103 V fluorescent light bulb (TL-D Standard Colours; Philips & Co., Eindhoven, The Netherlands) as a light source at room temperature (28 °C) for four to seven days until sufficient hyphal growth was observed on the cover slip. For microscopic observation, a drop of lactophenol cotton blue (Sigma-Aldrich, St. Louis, MO, USA) was applied to the fungal mycelia on the cover slip and the cover slip was placed face-down onto a microscope slide. The length and width of 30 randomly selected macroconidia were measured to determine the mean length and width for each isolate. Descriptions of the morphological characteristics were adopted from the *Fusarium* Laboratory Manual [57].

### 2.3. DNA Extraction and PCR Amplification

Fungal genomic DNA was extracted as previously described [46], and subsequently used for polymerase chain reaction (PCR) amplification of *TEF1-α* and *RPB2* using previously described primers [22], listed in Appendix A. The PCR mixture (20 μL) contained 1× GoTaq G2 Green Master Mix (Promega, Madison, WI, USA), 0.8 μM primers, and 10 ng template DNA. DNA amplification was performed with an initial denaturation of 1 min at 95 °C followed by 35 cycles of 30 s at 95 °C, 60 s at 55 °C, and 90 s at 72 °C and a final extension of 5 min at 72 °C. PCR products were purified using the QIAquick PCR Purification kit (QIAGEN Inc., Valencia, CA, USA) following the manufacturer’s instructions.

### 2.4. DNA Sequencing and Analysis of TEF1-α and RPB2 Sequences

DNA sequencing of PCR products was carried out by First BASE Laboratories Sdn Bhd (now known as Apical Scientific Sdn Bhd, Seri Kembangan, Malaysia). The *TEF1-α* and *RPB2* sequences were compared with other fungal DNA sequences using the online NCBI BLAST tool (https://blast.ncbi.nlm.nih.gov/Blast.cgi; 27 February 2022). For *Fusarium* species identification, the *TEF1-α* and *RPB2* sequences were used in a BLAST search against the *Fusarium* MLST database (https://fusarium.mycobank.org/page/Fusarium_identification; 27 February 2022). The accession numbers for the *TEF1-α* and *RPB2* sequences are listed in Table 1.

### 2.5. Phylogenetic Analyses

A representative dataset of *TEF1-α* and *RPB2* sequences of FSSC clades 1, 2, and 3 species were extracted from the GenBank DNA repository (Appendix A). The sequences were selected to best represent the global distribution of clinical and environmental isolates of the FSSC. The phylogenetic relationships between the concatenated *TEF1-α* and *RPB2* sequences were investigated with two independent methods: Maximum Parsimony (MP) and Maximum Likelihood (ML). MP analysis was performed using MEGA 10.0.5 [58]. Trees were generated for 100 replicates of random stepwise addition of sequences and the Subtree-Pruning-Regrafting (SPR) algorithm [59], with all characters given equal weight. Branch support for the MP analysis was estimated by performing 1000 bootstrap replicates with a heuristic search of 10 random–addition replicates for each bootstrap replicate [60]. Phylogenetic trees were visualized using the program Treeview [61]. ML analysis was performed using RAxML-HPC2 version 8.2.12 on XSEDE [62] via the CIPRES Science Gateway Portal version 3.1 [63]. The best evolutionary model TIM2+I+G4 for the *TEF1a-RPB2* dataset was calculated using ModelTest-NG on XSEDE [64].

### 2.6. Fusarium Growth Characteristics

Mycelial dry weights were used to determine the growth behaviour of selected isolates. Using optical densities to measure growth was not possible, because *Fusarium* cells grew as large, fluffy balls of mycelia after ~15 h incubation in liquid medium.

Inoculum suspensions of microconidia were prepared by adding 5 mL sterile saline to 5-day old PDA plate cultures, gently resuspending the cells by scraping with the end of a 1 mL pipette tip, and filtering the cell suspension through a two-layered cheesecloth. The cells were harvested in a 15 mL centrifuge tube by centrifugation for 5 min at 8000× *g* and resuspension of the cell pellet in the required volume of potato dextrose broth (PDB), containing 20% glycerol, to adjust the cell suspension to OD_530_ = 10 (~1 × 10^8^ cfu/mL). The microconidia suspension was aliquoted in 1.5 mL microcentrifuge tubes and used immediately for inoculation or stored at −80 °C until required.

The growth characteristics of *F. keratoplasticum* Fk2309 and FkDir61, *F. falciforme* Ff4290 and *F. striatum* Fstr541 were determined for two media: PDB and RPMI-1640 (R6504; Sigma-Aldrich, St. Louis, MO, USA) supplemented with 2% glucose (i.e., RPMI-1640 supplemented with 18 g glucose). Ten microlitre cell suspensions (i.e., 10^6^ cfu) of microconidia were used to inoculate flasks containing 50 mL medium, one for each time point of harvest. Inoculated media were incubated at 28 °C with shaking at 200 rpm. The cell suspensions were harvested by filtration through a glass fibre filter (No. 6; Schleicher & Schuell BioScience GmbH, Dassel, Germany) using a vacuum pump. The filters were dehydrated in a 30 °C incubator for 24 h and the cell dry weight was determined by subtracting the dry weight of the same filter measured before cell harvest.

## 3. Results

### 3.1. Molecular Identification

The 30 FSSC isolates from Malaysia were designated to species using their *TEF1-α* and *RPB2* sequences and the polyphasic identification system of the *Fusarium* MLST database (Table 1). The 30 FSSC isolates comprised thirteen *F. keratoplasticum* (FSSC 2), six *F. suttonianum* (FSSC 20), five *F. falciforme* FSSC 3+4), two *F. cyanescens* (FSSC 27), and one isolate each of *F. petroliphilum* (FSSC 1), *F. waltergamsii* (FSSC 7), *Fusarium* sp. (FSSC 12) and *F. striatum* (FSSC 21). Most (11; 85%) *F. keratoplasticum* strains were clinical isolates as were the single isolates of *F. petroliphilum* and *F. striatum* while most (5; 83%) *F. suttonianum* strains and the single isolates of *F. waltergamsii*, *Fusarium* sp. of FSSC 12 and *F. cyanescens* were of environmental origin.

An ML phylogenetic tree for the concatenated *TEF1-α* and *RPB2* sequences (2359 nucleotides) of 62 selected FSSC isolates from across the globe, including the 30 isolates from Malaysia, is presented in Figure 1. The majority (23 out of 30) of the Malaysian FSSC isolates (bold in Figure 1) belonged to three closely related FSSC species: *F. keratoplasticum* (FSSC 2), *F. falciforme* (FSSC 3+4) and *F. suttonianum* (FSSC 20). Isolates of the FSSC 2 or the FSSC 20 lineage could be clearly distinguished with the base-root branch of each showing 100% bootstrap support for both the MP and ML algorithms (Figure 1). However, the assignment of *F. falciforme* isolates to the FSSC 3+4 lineage was less clearly defined, with the base-root branch showing only 90% MP and 94% ML bootstrap support. One of the 30 Malaysian isolates (Fp667) was placed in the FSSC 1 lineage of *F. petroliphilum*. The *F. striatum* isolate Fstr541, a human blood isolate, formed a distinct branch (100% bootstrap support) with two closely related *F. striatum* sequences, one an insect-isolate from Columbia (NRRL52699) and the other an environmental isolate from Panama (NRRL22101; Figure 1). The sub-division of isolates within the FSSC 2 and FSSC 3+4 lineages (e.g., *F. keratoplasticum* FSSC 2-a, -f, -h, -k) mostly followed their haplotype assignment [22], although with less convincing bootstrap support for the individual sub-groups (Figure 1).

### 3.2. Morphological Characteristics

To investigate whether morphological characteristics correspond with the molecular species identification, we examined the macroscopic and microscopic characteristics of 25 of the 30 FSSC isolates (Table 1). The species of individual FSSC isolates could not be identified with certainty based on their colony morphology alone (Figure 2A,B). The macroconidia morphologies (Figure 3) provided more distinctive species-specific characteristics, although these too were imprecise and crossed species boundaries.

#### 3.2.1. Colony Morphology

The growth rate of colonies grown on PDA for nine days at 28 °C with 12 h light (day) and 12 h (night) dark periods ranged from 6.1 to 9.4 mm (diameter)/day (Table 2). Early hyphae were hyaline and sparse, radiating out from the centre of the agar plug containing the mycelial inoculum. The aerial mycelia of all young colonies (3–5 days) were white, but their colour and appearance changed with age resulting in rings of white interspersed by coloured patches ranging from green to yellow, orange, brown and dark brown (see images labelled (a) in Figure 2A,B). The colour of the colony after two weeks of growth, viewed from underneath, ranged from shades of yellow to dark brown (see images labelled (b) in Figure 2A,B). The mean colony diameters on day-9 incubation for the ten *F. keratoplasticum*, four *F. falciforme* and five *F. suttonianum* isolates were 63 ± 8.39, 75 ± 3.40, and 81 ± 4.41 mm, respectively (Table 2).

#### 3.2.2. Microconidia and Chlamydospores

Microconidia were present in all 25 FSSC isolates examined, with either single oval shaped cells or two cells separated by a septum and reniform (kidney-shaped). Representative examples for isolates of each of the eight FSSC species are presented in the images labelled (b) in Figure 3. Chlamydospores, images labelled (c) in Figure 3, were only observed in 14 of the 25 isolates: two *F. falciforme* (Ff4290, Ff4324), three *F. suttonianum* (Fs3769, Fs3784, Fs3924), seven *F. keratoplasticum* isolates (Fk553, Fk994, Fk2309, Fk2353, Fk2622, FkDI17, FkDir61), one each of *F. waltergamsii* (FwgDE4) and *Fusarium* sp. (FspDE40). When present, they were all globose and terminal rather than intercalary, and existed singly, paired, or in chains. *F. petroliphilum*, *F. keratoplasticum*, and *F. falciforme* had smooth-walled chlamydospores while *F. suttonianum* had rough-walled chlamydospores (Figure 3). Chlamydospores were absent in Fstr541 and FcDir16 when grown on CLA medium.

#### 3.2.3. Macroconidia

As expected, the macroconidia morphology was the most distinctive feature for the identification of *Fusarium* species [57]. All four *F. falciforme* isolates, but only about half of the *F. keratoplasticum* and *F. suttonianum* isolates, had macroconidia with at least 2 septa (Table 2). None of the ten *F. keratoplasticum* or the four *F. falciforme* isolates, and only two *F. suttonianum* isolates, had macroconidia with 4 or 5 septa (Table 2). Yet, all isolates of the remaining five species, *F. petroliphilum*, *F. striatum*, *F. waltergamsii*, *Fusarium* sp., and *F. cyanescens* had macroconidia with both 2 or 3 and 4 or 5 septa (Table 2). The mean lengths and widths of the macroconidia for isolates are presented in Table 2. The mean lengths ranged from 21–42 µm for 2 or 3 septa, and 31–51 µm for 4 or 5 septa (Table 2). Although there was a clear difference in the average length of the macroconidia with 2 or 3 septa between the five *F. keratoplasticum* (22.43 ± 1.60 µm), the four *F. falciforme* (25.70 ± 3.51 µm), and the three *F. suttonianum* isolates (27.77 µm ± 4.69; Table 3), inter-species differences were not statistically significant; there was too much of an overlap between individual isolates to use macroconidia length to assign isolates to a particular species (Table 2).

Representative images of macroconidia for the eight species identified in this study, *F. petroliphilum*, *F. keratoplasticum*, *F. falciforme*, *F. waltergamsii*, *Fusarium* sp., *F. suttonianum*, *F. cyanescens* and *F. striatum*, are presented in Figure 3 (labelled (a)). The shape of the macroconidia of all eight species was straight or slightly curved. *F. petroliphilum* macroconidia were much more abundant than for any of the other FSSC species investigated. *F. petroliphilum* macroconidia were elongate and slightly curved in shape and most contained 4 to 5 septa. The triangular-shaped apical and basal cells were blunt and barely notched, respectively. *F. keratoplasticum* macroconidia were straight and cylindrical to slightly curved with up to 3 septa for 50% (5/10) of the isolates. Similar to *F. petroliphilum*, the triangular shaped apical cells were blunt, and the basal cells were barely notched.

The macroconidia of the *F. keratoplasticum* isolates were also noticeably shorter than for most other isolates (Table 2 and Table 3). All four *F. falciforme* isolates only had macroconidia with 2–3 septa. They were slightly curved and larger than the *F. keratoplasticum* macroconidia, but they possessed similar triangular-shaped blunt apical and barely notched basal cells. *F. waltergamsii* macroconidia had distinctly hooked apical cells and notched to foot-like basal cells. *Fusarium* sp. of the FSSC 12 had macroconidia with slightly straighter dorsal, straight apical and barely notched basal cells. *F. suttonianum* macroconidia were somewhat straight on both dorsal and ventral lines, with moderate curvature that was more prominent in the apical and basal cells with 4–5 septa. The *F. striatum* macroconidia with 2 or 3 septa were slightly thinner (2.44 ± 0.47 µm; Table 2), straight, and almost needle-like whereas the widths for most other isolates ranged from 2.79 µm for Fp667 to 3.53 µm for Ff4290, most being ~3 µm in width (Table 2). The *F. striatum* macroconidia had hooked apical and barely notched, foot-shaped basal cells (Figure 3). *F. cyanescens* macroconidia were slightly curved, with hooked apical cells and barely notched basal cells.

Seven (Fk620, Fk553, Fk0168, Fk2622, FkDir61, Fs3784, Fs4279) of the 25 isolates grown on CLA did not produce any macroconidia (i.e., 5 of 10 *F. keratoplasticum*, equally distributed among clinical and environmental isolates, and 2 of 5 *F. suttonianum* isolates; Table 2). The remaining 5 *F. keratoplasticum* isolates (Fk2781, Fk2309, Fk2353, Fk994, FkDI17) only produced macroconidia with up to 3 septa, as did all 4 *F. falciforme* isolates (Ff4225, Ff4290, Ff4324, Ff4325). Only 8 of the 25 isolates (Fp667, Fstr541, FwgDE4, FspDE40, Fs3769, Fs3783, FcDir16, FcDir23) produced macroconidia with 4 or 5 septa. They included all isolates of *F. petroliphilum* (1), *F. striatum* (1), *F. waltergamsii* (1), *Fusarium* sp. (1) and *F. cyanescence* (2) and 2 of the 5 *F. suttonianum* isolates (Table 2 and Figure 3).

### 3.3. Growth of F. keratoplasticum, F. striatum and F. falciforme in PDB and RPMI

Mycelial dry weight measurements were used to create growth curves for *F. keratoplasticum* Fk2309 and FkDir61, *F. striatum* Fstr541, and *F. falciforme* Ff4290. The generation time (Td) for all four strains in PDB was ~ 3 h while their generation time in RPMI ranged from 4.3 h to 6.6 h with FkDir61 being the slowest growing strain (Figure 4). An extrapolation of the exponential growth lines towards the initial inoculum size of <0.5 mg/50 mL provided an estimate for the lag phase of each strain (Figure 4). All four strains took extended periods of time (they each had a ~10–15 h lag phase) to adapt to growth in PDB liquid medium, even though the microconidia inoculum was prepared from cells grown on PDA solid medium. Although all four *Fusarium* isolates grew much slower (~1.5–2 times slower) in RPMI, the estimated lag phases were surprisingly short (~7 h) for Fstr541 and Ff4290 and almost non-existent for Fk2309 and FkDir61 (<1–3 h) (Figure 4). All four isolates reached much higher (~3 times) cell densities (~100 mg/50 mL) in stationary phase in PDB than in RPMI (~32 mg/50 mL), except for Ff4290 which reached similarly low cell densities of ~32 mg/50 mL in both media (Figure 4). Based on these results, mid-logarithmic phase is reached after 21 h incubation in PDB.

## 4. Discussion

Invasive fusariosis is a severe disease of the immunocompromised, mostly those with hematologic malignancies [36]. Disseminated fusariosis is usually confirmed by a positive blood culture and has a poor prognosis with a high mortality rate [11]. Only two (Fk0168 and Fstr541) of the 15 clinical isolates were from positive blood cultures. The patients’ underlying conditions were, unfortunately, not recorded. The remaining 13 clinical FSSC isolates were from superficial infection sites; eight were from nail, two from skin, and three from eye infections. Thus, the majority (87%; 13 of 15) of the fusariosis cases were superficial nail (8 *F. keratoplasticum* isolates), eye (3; one isolate each of *F. petroliphilum*, *F. falciforme* and *F. suttonianum*), and skin infections (2 *F. keratoplasticum* isolates) similar to previous reports [11]. The species distribution of the 15 clinical FSSC isolates agreed with previous reports that found *F. keratoplasticum* (FSSC 2) to be one of the FSSC species most frequently isolated in the clinic [21,65].

Three of the eight FSSC species contained both environmental and clinical isolates: *F. keratoplasticum*, *F. falciforme*, and *F. suttonianum* (Table 1). *F. keratoplasticum* (FSSC 2) was the most prevalent clinical isolate (73%; 11 of 15), all but one (blood isolate Fk0168; Table 1) were skin or nail isolates, in good agreement with previous reports [21,65,66,67,68,69,70]. *F. keratoplasticum* is also an important veterinary pathogen, causing infections in equine and marine vertebrates as well as in invertebrates [71]. In contrast, the majority of the *F. falciforme* (80%; 4 of 5) and *F. suttonianum* (83%; 5 of 6) isolates were of environmental origin. Ff0020 and Fs263 were the only two isolates of human origin (eye; Table 1).

The other five FSSC species were isolated at a much lower frequency. They were either clinical (*F. petroliphilum* and *F. striatum*, one each) or environmental isolates (*F. waltergamsii* and *Fusarium* sp., one each, and two *F. cyanescens* isolates; Table 1). The *F. suttonianum* isolates accounted for 33% (5 of 15) of all environmental isolates. Others have also recovered *F. suttonianum* isolates from skin, nail, corneal ulcer, and blood samples [25]. The *F. falciforme* isolates accounted for ~27% (4 of 15) of the environmental isolates, and, like *F. suttonianum*, only one was of clinical origin (eye). Interestingly, *F. falciforme* has been reported as the predominant species isolated in South India accounting for 83% [72] and 93% [73] of all FSSC isolates. The only *F. petroliphilum* (FSSC 1) isolate was of clinical origin (eye)*. F. petroliphilum* isolates have been previously obtained from air samples which caused airborne transmission of IF [74] and even fatal disseminated fusariosis [75].

*F. suttonianum* appears innately resistant to most azole antifungals. Based on the azole susceptibility data that we previously reported [46], the six *F. suttonianum* isolates in this study were resistant to itraconazole, posaconazole, and also voriconazole, and all carried the 23 bp *CYP51A* promoter deletion that was tightly associated with voriconazole resistance. *F. suttonianum* resistance to other triazoles including the agricultural fungicides tebuconazole and propiconazole has also been reported [66]. The voriconazole resistant *F. keratoplasticum* isolates Fk2309, Fk2781, and FkDI17 also carried the 23 bp *CYP51A* promoter deletion (Figure 1 and [46]) which suggested cross-species DNA exchange between their common ancestors. The fact that the voriconazole resistant *F. suttonianum* and *F. keratoplasticum* isolates were of both environmental and clinical origin (Figure 1) supports the hypothesis that voriconazole resistance was selected for by fungicide use in agriculture. Whether the two additional voriconazole resistant isolates, Ff0020 and Fstr541 (marked with an asterisk in Figure 1), also contain the 23 bp *CYP51A* deletion remains to be investigated. It is quite likely that the rather distantly related isolate Fstr541 (Figure 1) and possibly also the only voriconazole resistant *F. falciforme* isolate Ff0020, both of clinical origin, have different voriconazole resistance conferring mutations. One possibility may be gain of function mutations in a transcription factor that causes the overexpression of orthologs of the recently discovered multidrug efflux pump, *F. keratoplasticum* Abc1 [76].

The identity of 25 of the 30 FSSC isolates was initially assessed with traditional identification methods by carefully analysing the macroscopic and microscopic features of cell cultures grown on CLA medium [77]. The shape and size of the microconidia and clamydospores was rather uniform and provided no distinctive power for accurate species identification. However, the shape and size of the macroconidia provided more distinctive features. The presence of larger macroconidia with 3–5 septa, the shape of their spores and their apical and basal cells, the number of septa and the size of the macroconidia are key characteristics for the identification of FSSC species [77]. The macroconidia of most FSSC isolates (72%; 18 of 25) had 2 or 3 septa and only eight of them also had macroconidia with 4 or 5 septa (Table 2). A previous study of Malaysian FSSC isolates [78] reported similar results for four *F. keratoplasticum* and four *F. falciforme* isolates, although there were notable differences. All *F. keratoplasticum* and *F. falciforme* isolates had macroconidia with 3 or 4 septa and, as in our study, none had 5 septa. However, their average macroconidia size (length × width) [78] was ~50% greater (33.75 × 4.88 µm and 41.75 × 5.10 µm, respectively) than the average *F. keratoplasticum* and *F. falciforme* macroconidia sizes determined in the present investigation (Table 2 and Table 3). These differences are likely due to the different growth conditions used by the different laboratories. Cheri et al. (2015), for instance, incubated the CLA plate cultures at room temperature (25 ± 2 °C) rather than at 28 °C. Another study [79] investigated the morphology of *F. keratoplasticum* and *F. petroliphilum* isolates grown at 22 °C with alternating 12 h light/dark cycles using a 120 V UV light bulb as a light source instead of a 103 V cool white fluorescent light bulb. And a third study [25] of five FSSC species (FSSC 6, 7, 9, 20 and 43) used synthetic-nutrient-agar [80] with and without carnation leave pieces incubated at ‘room temperature’, a term that could vary significantly between different laboratories. Contrary to these studies [25,79,80], about half of the presently investigated *F. keratoplasticum* (5 of 10) and *F. suttonianum* (2 of 5) isolates had no apparent macroconidia at all. This could be due to subtle variations in the culturing conditions or a reflection of their different geographical origins.

The average colony diameters of isolates of the three major FSSC species *F. keratoplasticum*, *F. falciforme* and *F. suttonianum* grown on PDA for nine days at 28 °C were measured. However, as with any of the other morphological observations, there were strain variations that made it impossible to assign individual isolates to a particular FSSC species based on these measurements. There was also no other morphological characteristic that could distinguish between isolates of the various FSSC MLST sub-types (e.g., *F. keratoplasticum* FSSC 2-a, 2-f, 2-h; Figure 2A). One of the limitations of this study was the small number of FSSC isolates investigated which naturally precludes an accurate and representative description of the morphology of individual species.

We also investigated the growth characteristics of *Fusarium* isolates in PDB and RPMI media frequently used to cultivate fungi (PDB) or determine the drug susceptibilities (RPMI) of clinical and environmental fungal isolates. For fair comparisons, RPMI was adjusted to 2% glucose. The *Fusarium* isolates chosen included two clinical isolates from nail (Fk2309) and blood (Fstr541) and two environmental isolates (FkDir61, Ff4290). The two *F. keratoplasticum* isolates represented a voriconazole resistant (Fk2309) isolate with the 23 bp *CYP51A* promoter deletion and a voriconazole sensitive isolate with the wild type *CYP51A* promoter. All four strains grew equally fast in PDB with a ~3 h generation time. Their growth was ~1.5- to 2-times slower in RPMI (4.3–6.6 h). Interestingly, however, all strains appeared to have a shorter lag phase (~5–8 h) in RPMI, especially the two *F. keratoplasticum* isolates Fk2309 (~5 h) and FkDir61 (~7 h; Figure 4); the lag times for all four strains were longer (~10–15 h) when grown in PDB medium. A major difference between the two media is the pH. While the pH of PDB (pH 5.2) is designed for optimum fungal growth, RPMI (pH 7.2) mimics the growth conditions the fungus would encounter in humans. Whether the reduced lag times in RPMI or the inability to produce macroconidia with more than 3 septa by any of the *F. keratoplasticum* and *F. falciforme* isolates have anything to do with their increased likelihood to infect human tissue remains to be investigated.

The *TEF1-α* and *RPB2* sequences alone were not enough to correctly identify one of the 30 FSSC isolates, Fstr541. This isolate was initially identified as *F. falciforme* Ff541 (MLST 3+4 oo) [46]. Only the phylogenetic relationship of the 30 FSSC isolates with 35 closely related published FSSC sequences could identify this isolate as *F. striatum*. The concatenated *TEF1-α* and *RPB2* sequences were also not able to clearly separate isolates of different sub-lineages (e.g., 2-a, -d, -f, -h, -k or 3+4-k, -ii, -tt, -oo) for the two largest and rather diverse FSSC species, *F. keratoplasticum* and *F. falciforme* (Figure 1). *F. falciforme* isolates accounted for approximately one-third (63) of 191 unique haplotypes (i.e., FSSC 3+4-a to -kkk) of clade 3 of the FSSC in a previous investigation [22]. Previous studies used additional sequences for FSSC species identification; i.e., the entire internal transcribed spacer region (ITS) plus the D1+D2 fragments of the large rRNA subunit (LSU) with [81] or without [22] the calcium-binding messenger protein gene, calmodulin (*CAM*). LSU and ITS, however, were the least informative sequences to distinguish between closely related FSSC species [16,22]. This is why we did not use them in the present study. However, our study clearly demonstrates the need for additional housekeeping gene sequences such as *RPB1* and *TUB2* to distinguish individual *F. keratoplasticum* and *F. falciforme* isolates more clearly from each other. The *Candida albicans* MLST scheme, for instance, uses seven housekeeping genes (*AAT1a*, *ACC1*, *ADP1*, *MPIb*, *SYA1*, *VPS13*, and *ZWF1b*) [82]. The MLST schemes of several other pathogenic *Candida* species (*C. krusei*, *C. tropicalis*, *C. glabrata, and C. dubliniensis*) use at least six gene sequences, eight sequences are used for *C. dubliniensis* [83], and the *A. fumigatus* MLST scheme uses seven gene sequences (*ANXC4*, *BGT1*, *CAT1*, *LIP*, *MAT1-2*, *SODB*, and *ZRF2*) [84] to accurately assign isolates to the various geographically-, epidemiologically-, or specific host-associated sub-lineages.

## 5. Conclusions

*F. keratoplasticum* isolates were by far the most frequent clinical (11 of 15) isolates. *F. suttonianum* (5 of 15) and *F. falciforme* (4 of 15) were the most frequent environmental isolates. The remaining five FSSC species were only isolated once (*F. petroliphilum*, *F. striatum*, *F. waltergamsii*, *Fusarium* sp.) or twice (*F. cyanescens*). Interestingly, none of the *F. keratoplasticum* or *F. falciforme* isolates investigated (10 and 4, respectively) produced macroconidia with 4 or 5 septa, and only about half of the *F. keratoplasticum* and *F. suttonianum* isolates produced any macroconidia at all. Yet, all isolates of the other five FSSC species produced macroconidia with 4 or 5 septa. None of the morphologic or microscopic observations were sufficient for the species level identification of any of the FSSC isolates. Far more accurate results were obtained with phylogenetic analysis using the *TEF1-α* and *RPB2* sequences. However, the inclusion of additional gene sequences is recommended for a more accurate resolution of their phylogenetic relationships, especially for the two most diverse species *F. keratoplasticum* and *F. falciforme*, which are also the two main FSSC species causing life-threatening IF.

## Figures and Tables

**Figure 1 jof-08-00845-f001:**
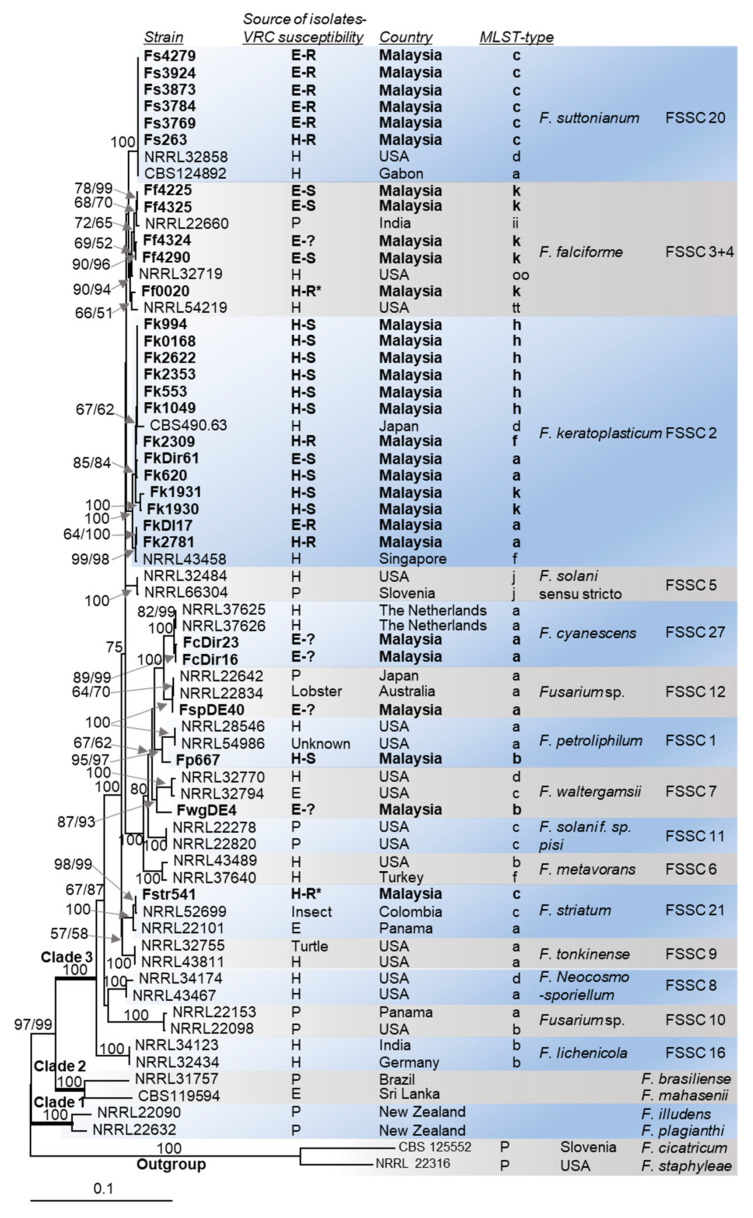
Maximum likelihood (ML) phylogram of the concatenated *TEF1*-α and *RPB2* sequences for the 30 Malaysian FSSC isolates and 37 publicly available sequences of globally distributed FSSC isolates. DNA sequences of two close relatives (*F. cicatricum* and *F. staphyleae*) were used as the outgroup. Isolates characterised in this study are highlighted in bold font. The source of individual isolates (E = environmental, H = hospital, P = plant) is shown in the column to the right of the strain identifier. The voriconazole resistant (MIC > 32 mg/L) isolates with the 23 bp *CYP51A* promoter deletion and the voriconazole susceptible (MIC ≤ 12 mg/L) isolates are indicated with R and S, respectively. The voriconazole susceptibilities of isolates marked with a question mark (?) were not determined. Technical issues prevented the confirmation of the 23 bp *CYP51A* promoter deletion in the voriconazole resistant *F. falciforme* (Ff0020) and *F. striatum* (Fstr541) isolates (marked with asterisks; *). The roots, with 100% bootstrap support, of the three major FSSC clades (clades 1, 2 and 3) are in bold. Numbers at internodes represent the percentage maximum parsimony (MP) and maximum likelihood (ML) bootstrap support (MP-BS/ML-BS) of 1000 replicates; a single value means that both values were identical. The scale bar indicates the number of nucleotide substitutions per position.

**Figure 2 jof-08-00845-f002:**
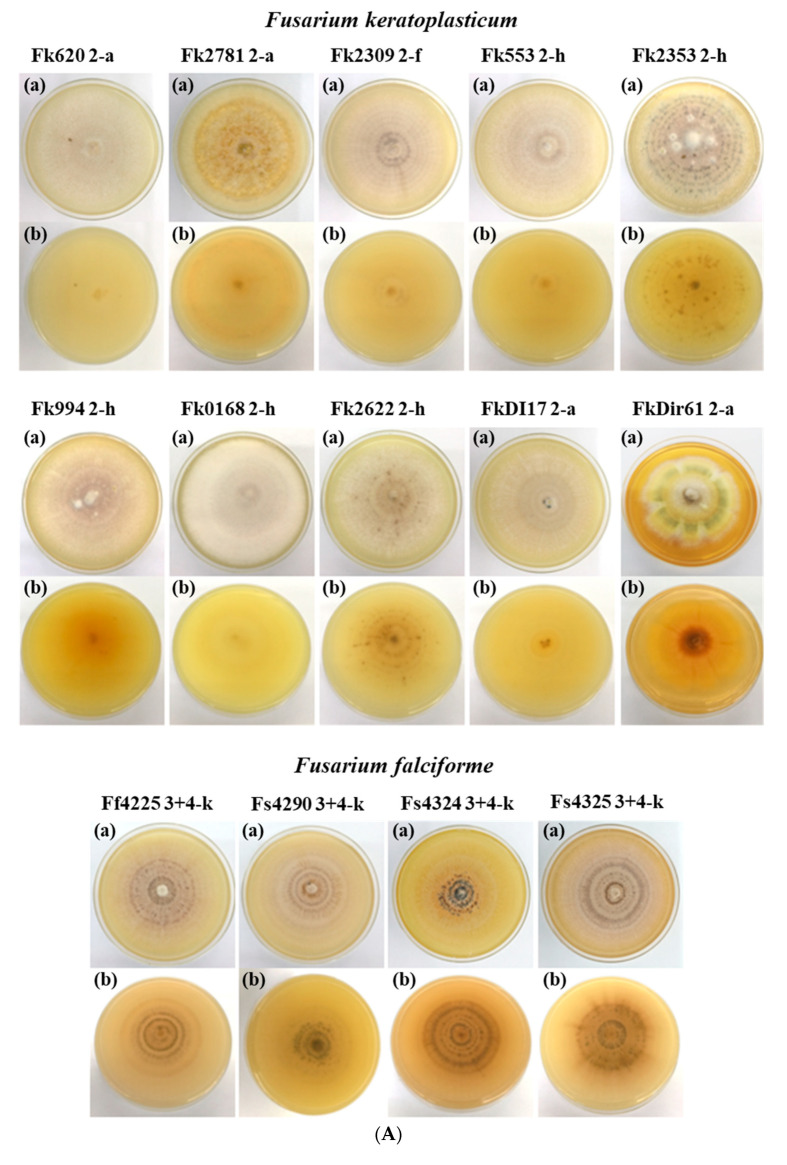
(**A**). Colony morphologies of *F. keratoplasticum* (ten) and *F. falciforme* (four) isolates grown on PDA at 28 °C for two weeks with alternating 12 h light and 12 h dark periods. View from the top (a) and underneath (b). (**B**). Colony morphologies of *F. suttonianum* (five), *F. cyanescens*. (two), and one each of *F. petroliphilum*, *F. striatum*, *F. waltergamsii* and *Fusarium* sp. isolates grown on PDA at 28 °C for two weeks with alternating 12 h light and 12 h dark periods. View from the top (a) and underneath (b).

**Figure 3 jof-08-00845-f003:**
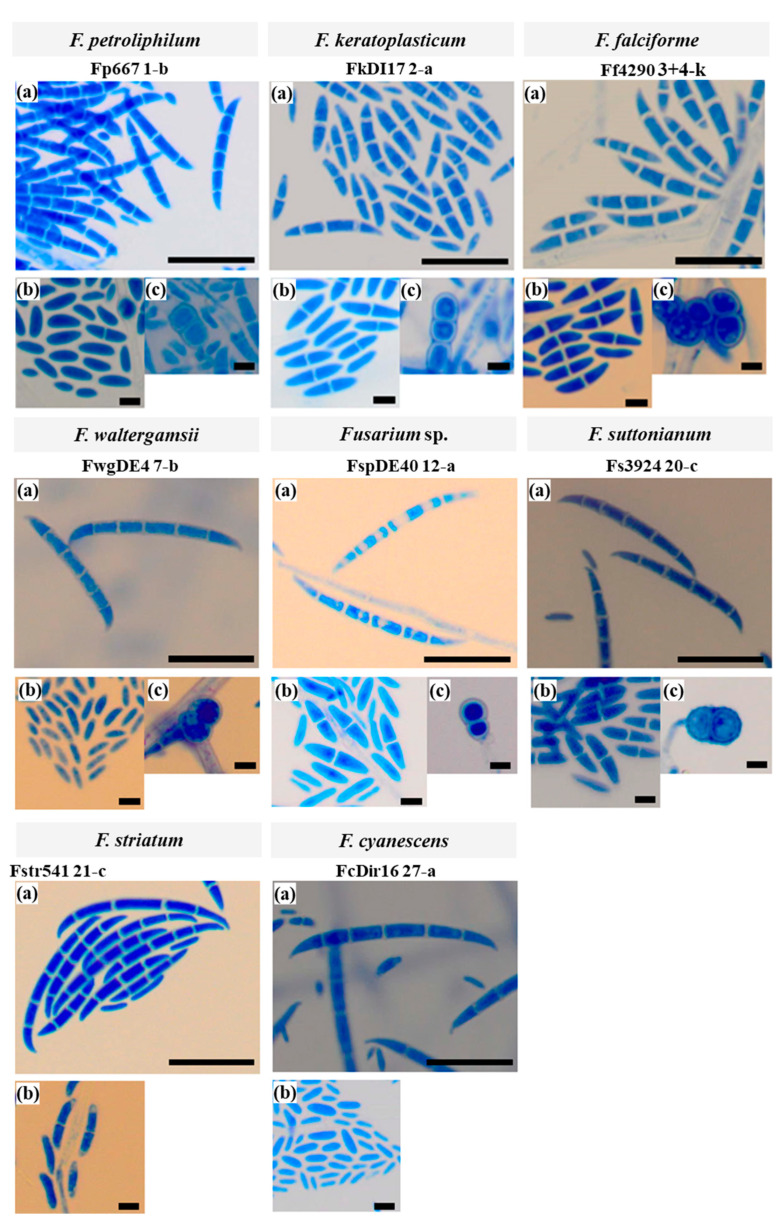
Microscopic images at 40× magnification of eight FSSC isolates representing the eight species identified among the collection of the 30 Malaysian FSSC isolates. Cells were grown on CLA at 28 °C for four to seven days with alternating 12 h light and 12 h dark periods. Cells were stained with lactophenol cotton blue. Macroconidia (a; scale bar: 100 µm), microconidia (b; scale bar: 20 µm), and chlamydospores (c; scale bar: 20 µm).

**Figure 4 jof-08-00845-f004:**
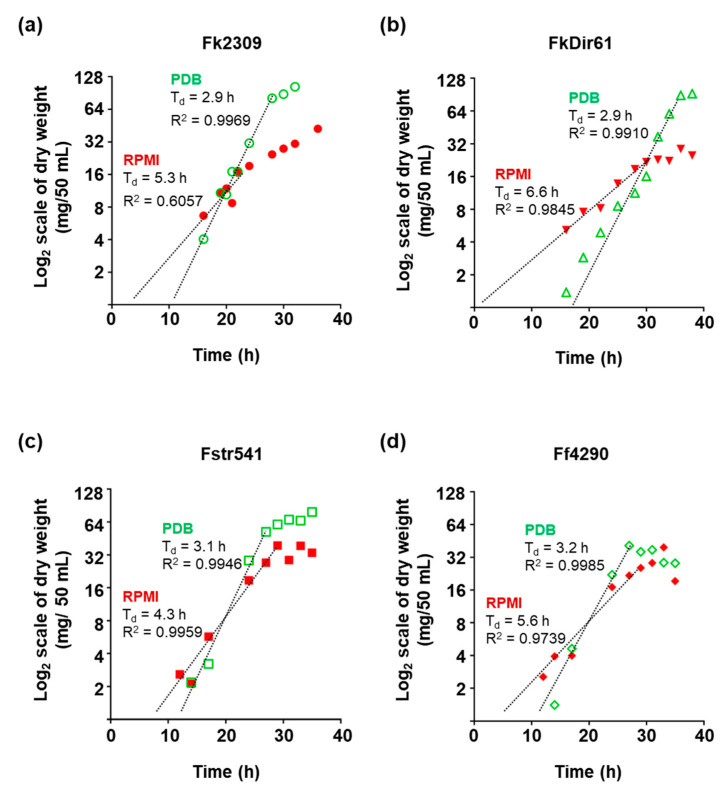
Growth curves of two *F. keratoplasticum* strains, Fk2309 (**a**) and FkDir61 (**b**), and of *F. striatum* and *F. falciforme* isolates Fstr541 (**c**) and Ff4290 (**d**) grown in PDB (green circles) or RPMI (red circles) medium, respectively. Dotted lines are the trendlines for the exponential growth phase that were used to calculate the generation time (T_d_). The results presented were from a single experiment.

**Table 1 jof-08-00845-t001:** List of 15 clinical and 15 environmental FSSC isolates from Malaysia, their MLST type and their GenBank accession numbers for *TEF1-α* and *RPB2*.

Isolate	Source ^1^	Species	MLST Type ^2^	GenBank Accession Number	
				*TEF1-α*	*RPB2*
**Clinical Isolates**					
Fp667	Eye	*F. petroliphilum*	1-b	MN178239	MN263125
Fk620	Skin	*F. keratoplasticum*	2-a	MN178238	MN263124
Fk2781	Nail	*F. keratoplasticum*	2-a	MN178234	MN263120
Fk2309	Nail	*F. keratoplasticum*	2-f	MN178231	MN263117
Fk553	Skin	*F. keratoplasticum*	2-h	MN178237	MN263123
Fk2353	Nail	*F. keratoplasticum*	2-h	MN178232	MN263118
Fk994	Nail	*F. keratoplasticum*	2-h	MN178240	MN263126
Fk0168	Blood	*F. keratoplasticum*	2-h	MN178228	MN263114
Fk2622	Nail	*F. keratoplasticum*	2-h	MN178233	MN263119
Fk1049 *	Nail	*F. keratoplasticum*	2-h	MN178241	MN263127
Fk1931 *	Nail	*F. keratoplasticum*	2-k	MN178230	MN263116
Fk1930 *	Nail	*F. keratoplasticum*	2-k	MN178229	MN263115
Ff0020 *	Eye	*F. falciforme*	3+4-k	MN178227	MN263113
Fstr541	Blood	*F. striatum*	21	MN178236	MN263122
Fs263 *	Eye	*F. suttonianum*	20-c	MN178235	MN263121
**Environmental isolates**					
FkDI17	Grass	*F. keratoplasticum*	2-a	MN178221	MN263107
FkDir61	Grass	*F. keratoplasticum*	2-a	MN178225	MN263111
Ff4225	Tobacco	*F. falciforme*	3+4-k	MN178212	MN263098
Ff4290	Straw compost	*F. falciforme*	3+4-k	MN178215	MN263101
Ff4324 **	Soil	*F. falciforme*	3+4-k	MN178216	MN263102
Ff4325	Honeydew	*F. falciforme*	3+4-k	MN178217	MN263103
FwgDE4 **	Soil	*F. waltergamsii*	7-b	MN178218	MN263104
FspDE40 **	Soil	*Fusarium* sp.	12-a	MN178220	MN263106
Fs3769	Coconut tree	*F. suttonianum*	20-c	MN178207	MN263093
Fs3784	Mangrove	*F. suttonianum*	20-c	MN178209	MN263094
Fs3873	Grass	*F. suttonianum*	20-c	MN178208	MN263095
Fs3924	Sugarcane	*F. suttonianum*	20-c	MN178210	MN263096
Fs4279	Dragon fruit	*F. suttonianum*	20-c	MN178214	MN263100
FcDir16 **	Soil	*F. cyanescens*	27-a	MN178223	MN263109
FcDir23 **	Soil	*F. cyanescens*	27-a	MN178224	MN263110

^1^ The body site or plant or soil environment from which the samples were collected. ^2^ The MLST was determined based on polyphasic identification using the *Fusarium* MLST database. Numbers were assigned to designate species and lowercase letters to identify unique haplotypes. * These isolates were excluded from the morphological investigations. ** These are additional isolates that were not included in the study by James et al. [46].

**Table 2 jof-08-00845-t002:** Morphological features of FSSC isolates.

Isolate ^1^	Mycelial Growth on PDA at 28 °C (Day-9)			Length and Width of Macroconidia (µm) ^2,3,4^			
	Top View	Viewed from Underneath	Colony Diameter (mm)	2 and 3 Septa		4 and 5 Septa	
				Length	Width	Length	Width
**CLINICAL ISOLATES (10)**							
Fp667	white, brown	brown, yellow	55	27.10 ± 3.65	2.79 ± 0.41	30.61 ± 2.68	2.99 ± 0.39
Fk620	White	white	70	none	none	none	none
Fk2781	white brownish	yellow	66	23.68 ± 2.95	3.05 ± 0.52	none	none
Fk2309	white with purple stripes	white	55	21.37 ± 2.56	3.04 ± 0.47	none	none
Fk553	White	yellow	66	none	none	none	none
Fk2353	white with purplish dark spots	white	60	20.85 ± 1.93	3.02 ± 0.46	none	none
Fk994	white, brown	orange	63	21.71 ± 3.22	2.91 ± 0.49	none	none
Fk0168	White	yellow	68	none	none	none	none
Fk2622	White	white	58	none	none	none	none
Fstr541	dark brown, white	dark brown, yellow	65	26.63 ± 3.47	2.44 ± 0.47	31.49 ± 9.90	2.81 ± 0.49
**ENVIRONMENTAL ISOLATES (15)**							
FkDI17	White	yellow	78	24.56 ± 1.92	2.95 ± 0.50	none	none
FkDir61	white, dark brown, yellow, green	dark brown, orange, yellow	46	none	none	none	none
Ff4225	white, dark brown	dark brown, orange, yellow	78	21.50 ± 1.64	3.37 ± 0.35	none	none
Ff4290	White	Yellow	76	25.51 ± 3.31	3.53 ± 0.53	none	none
Ff4324	white, purple	white, purple	78	25.82 ± 2.00	3.27 ± 0.51	none	none
Ff4325	white, brown	Brown	70	30.09 ± 2.71	3.07 ± 0.39	none	none
FwgDE4	white, brown	brown, yellow	75	30.68 ± 3.51	2.95 ± 0.52	40.56 ± 2.79	2.92 ± 0.45
FspDE40	white, dark brown	dark brown, orange	49	40.71 ± 2.15	2.98 ± 0.49	47.65 ± 4.27	2.94 ± 0.43
Fs3769	White	yellow	80	24.83 ± 3.04	3.17 ± 0.42	41.02 ± 2.91	3.29 ± 0.41
Fs3784	dark brown, white	dark brown, yellow	85	none	none	none	none
Fs3873	White	white	80	33.18 ± 2.14	3.30 ± 0.54	41.32 ± 2.43	3.07 ± 0.52
Fs3924	White	yellow	85	25.29 ± 5.02	3.31 ± 0.39	none	none
Fs4279	White	yellow	73	none	none	none	none
FcDir16	white, dark brown	dark brown, orange, yellow	63	41.56 ± 2.83	2.86 ± 0.41	50.76 ± 3.86	2.74 ± 0.39
FcDir23	White	white	63	29.85 ± 4.89	3.02 ± 0.51	36.71 ± 1.37	2.95 ± 0.36

^1^ Species abbreviation: *F. petroliphilum* (Fp), *F. keratoplasticum* (Fk), *F. striatum* (Fstr), *F. falciforme* (Ff), *F. waltergamsii* (Fwg), *Fusarium* sp. (Fsp), *F. suttonianum* (Fs), *F. cyanescence* (Fc). ^2^ Size of macroconidia grown on CLA for four to seven days at 28 °C. ^3^ Mean values of 30 random conidia ± standard deviation. ^4^ none means not present.

**Table 3 jof-08-00845-t003:** Average length and width of *Fusarium* species macroconidia with 2 or 3 septa. Values are presented as the means ± standard deviation.

Species	Macroconidia ^1^	
	Length (µm)	Width (µm)
*F. keratoplasticum* (*n* = 5) ^2^	22.43 ± 1.60	2.99 ± 0.06
*F. falciforme* (*n* = 4)	25.70 ± 3.51	3.32 ± 0.19
*F. suttonianum* (*n* = 3) ^3^	27.77 ± 4.69	3.26 ± 0.06

^1^ Size of macroconidia grown on CLA for four to seven days at room temperature, 28 °C. ^2^ Only five of the ten *F. keratoplasticum* isolates investigated (Table 1) had macroconidia with 2 or 3 septa. ^3^ Only three of the five *F. suttonianum* isolates (Table 1) had macroconidia with 2 or 3 septa.

## Data Availability

Data (DNA sequences) can be obtained freely from GenBank using the accession numbers provided or by contacting the first or corresponding author.

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
