# Peer review of "Morphology, Phenotype, and Molecular Identification of Clinical and Environmental *Fusarium solani* Species Complex Isolates from Malaysia"

_jof, 2022, doi:10.3390/jof8080845_

Round 1

Reviewer 1 Report

Morphology, phenotype and molecular identification of clinical and environmental Fusarium solani species complex isolates from Malaysia

The author use standard molecular and morphological methods to determine phylogenetic placement for some Malaysian Fusarium isolates. The article is well written, and the figures are well designed. The major issue is with the discussion as it repeats a lot of the results, but needs to compare the results in this manuscript to others. Since this paper is mostly morphology and growth, please compare these aspects to previously literature.

Other Line items

Line 211: How consistent was the weight of the glass fibre filter? As worded is seems like one filter was used to collect and another was used as the control weight. Could some of the changes in weight be due to differences in the two filter papers and not solely via fungal mass.

 Line 222-224: Its unusual to see citations in the result section, these sentences should be moved to the discussion where the current findings are related to previous findings.

 Line 239-241: This is sentence is not a result from this manuscript so please remove and add to the discussion section.

 Line 328-330: This information would be best suited for the methods section and removed from the results.

 Figure 4 should be prior to the Discussion.

 Overall: The discussion seems to repeat a lot of the results. The authors should discuss the major findings and context to what others found. Were the morphologies similar to other studies on these species or are these isolates morphologically distinct. There needs to be some comparison and not just description. In addition, many studies have shown that Fusarium isolates require multigene sequences to yield species level ID so please cite those.

Reviewer 2 Report

The manuscript addresses an interesting study but has already been explored in Malaysia https://www.ncbi.nlm.nih.gov/pmc/articles/PMC4356886/. One of the authors participated here and in the previous research. The main contribution of this study is the molecular identification of the analyzed strains, but this has been already published by the same group https://www.frontiersin.org/articles/10.3389/fmicb.2020.00272/full. There, the MLST code is given for the isolates, the taxonomy at the species level is established and even the entries for the gene bank are the same. The authors may claim that five new isolates were included in the present study, but this does not provide a significant advance on our current knowledge on this subject. Finally, being pathogenic organisms, it results obvious to include the virulence profile within the phenotypical characterization, an aspect not explored by the authors.

Reviewer 3 Report

As general comment the work is well written and designed with relevant results.

In general terms the topic of the article is interesting, the methodology is explicitly presented and the results reported are interesting.

The structure of the paper is correct.

In my opinion, the abstract is too general, please reframe.

The introduction chapter should end with a paragraph indicating the purposefulness of the conducted research. Authors should clearly define the purpose of the work and formulate research hypotheses.

Materials and method section is well described and correspond to the aim set out in the manuscript. The tables and figures clearly presenting the obtained results with their appropriate interpretation.

The statistical calculation methods used in the research make the obtained results reliable and provide a basis for drawing correct conclusions.

The figures clearly presenting the obtained results with their appropriate interpretation.

The references are sufficient and necessary.

The paper needs some editorial corrections.

I recommend the publication of this manuscript in the Journal of Fungi journal after minor revisions.

Reviewer 4 Report

In general, the manuscript was well presented and has important findings of interest to the scientific community. However, minor revision are needed:

line 45 delete "mould"

lines 299, 300 delete (in between hyphae), (at the top of hyphae), (in between

hyphae). Educated readers should understand the "scientific " terms

line 515 change "were" to "was"

Round 2

Reviewer 1 Report

The authors have addressed my concerns.

Reviewer 2 Report

After the clarification by the authors in the rebuttal letter, and the inclusion of those differentiative aspects within the manuscript, I think this work is suitable for publication.